# T-PhISH-Net: Temporally Consistent Underwater Image Enhancement via a Transformer-Based Extension of PhISH-Net

Bjørn Christian Weinbach[1] and Reza Arghandeh[2]

[1]Western Norway Research Institute
[2]Western Norway University of Applied Sciences

## Abstract

Underwater video feeds often suffer from flicker — erratic frame-wise color and illumination shifts caused by wavelength-dependent light attenuation and scattering in water [1, 2]. This instability can degrade the performance of vision systems for marine robotics and monitoring, which require consistent imagery over time. We address this with T-PhISH-Net, a novel underwater enhancement model that extends the single-image PhISH-Net framework [3] into a temporally consistent video enhancer. T-PhISH-Net processes a causal window of frames and introduces three key innovations: (1) a motion-magnitude input channel feeding per-pixel movement cues to the network, (2) a causal Transformer encoder with a learnable decay gate to weigh past frames' contributions, and (3) an adaptive loss weighting scheme that balances fidelity and coherence by learning the weights of spatial and temporal loss terms during training. Experiments on four underwater video datasets show that T-PhISH-Net produces state-of-the-art enhancement quality and significantly reduces flicker. As a result, it achieves higher no-reference image quality scores than prior methods. The model runs in real time on sequential input, enabling flicker-free underwater video feeds that can benefit applications in ROV/AUV operations, marine ecology surveys, and infrastructure inspection.

## 1 Introduction

Underwater images and videos often suffer from color casts, haze, and low contrast due to the physics of light propagation in water [2]. Longer wavelengths (reds) are absorbed quickly, while shorter blue–green wavelengths dominate, yielding characteristic bluish tones and reduced visibility at depth [1]. Enhancing underwater visuals is crucial for tasks like habitat monitoring, marine robotics, and object detection [4, 5]. However, most deep learning–based underwater image enhancement (UIE) methods operate on individual frames [6], ignoring temporal dynamics. When applied to video, naive frame-by-frame enhancement can lead to temporal inconsistency or *flicker*: successive frames may vary unpredictably in brightness or color, confusing human observers and automated trackers. This problem has been explored in general video processing (e.g. enforcing consistency in post-processing filters) [7, 8], and has only recently attracted attention in underwater vision. For example, Xie et al. introduced a large-scale underwater video benchmark (UVEB) and a baseline method (UVE-Net) that leverages inter-frame information [9]. Yet, such approaches often rely on optical-flow-based warping or temporal smoothing operations and do not incorporate underwater imaging physics.

In this work, we propose T-PhISH-Net, a Transformer-based extension of a physics-inspired enhancer (PhISH-Net [3]) designed for temporally coherent underwater video enhancement. Our goal is to produce flicker-free enhanced videos without sacrificing per-frame image quality or interpretability. T-PhISH-Net bridges two previously separate paradigms: physics-guided UIE (which uses models of light attenuation/backscatter for principled single-image restoration [2]) and temporal deep learning (which uses past frames to stabilize outputs [7, 8]). By combining these, our method ensures that enhancements remain consistent across frames even under dynamic lighting or motion, while still performing accurate color and contrast correction per frame.

The full manuscript provides complete methodology and evaluation details [10].

## 2 Proposed Method

**Contributions:** T-PhISH-Net introduces several novel components to achieve temporally stable underwater enhancement:

- **Motion-aware four-channel encoder:** We feed a per-pixel motion magnitude map as a fourth channel alongside the RGB input frames. This extra channel provides explicit cues about inter-frame changes, helping the network distinguish actual scene motion from noise or flicker and adjust its enhancement accordingly.

- **Causal Transformer with learnable decay:** A lightweight Transformer encoder operates causally on a sliding window of past frames. Self-attention across frames allows the model

to learn correspondences and stabilize temporal variations. We introduce a single learned decay factor $\alpha \in (0, 1)$ that modulates the contribution of older frame features (exponentially decreasing with frame age). This adaptive gating lets the network learn how quickly to "forget" old information, emphasizing recent context while still leveraging useful past cues.

- **Adaptive loss weighting:** We balance fidelity vs. consistency by making the loss weights for each objective learnable instead of fixed. Each spatial or temporal loss term is given a trainable weight (kept non-negative via softplus) with a small regularization toward an initial prior value. This adaptive weighting scheme allows the model to automatically find an optimal trade-off between per-frame enhancement quality and cross-frame coherence, eliminating manual tuning of loss weights.

Figure 1 illustrates the T-PhISH-Net architecture. The model processes a causal window of $T$ frames (e.g. $T = 5$) simultaneously. Each frame is passed through a PhISH-Net style encoder to extract multi-scale features, which are flattened and fed into the temporal Transformer for self-attention across frames. The Transformer outputs are then used by a decoder to reconstruct the enhanced current frame. A temporal consistency loss is applied between consecutive outputs to suppress flicker. Notably, our design avoids explicit optical flow computation or frame warping, unlike many video enhancement methods. Optical flow is often unreliable in underwater scenes due to moving particulates, caustic light patterns, and low-contrast textures. By forgoing flow and instead using attention (aided by the motion magnitude cue), T-PhISH-Net achieves stability in a purely learning-driven manner, making it robust under challenging underwater conditions.

## 3  Results and Discussion

We evaluate T-PhISH-Net on four underwater video datasets: *Brackish* [11], *MVK* [12], *UOT32* [13], and *VDD-C* [14]. We compare our method against standard frame-by-frame enhancement and a recent deep video enhancer. In particular, we include a classical per-frame approach (CLAHE histogram equalization), the single-image deep model PhISH-Net [3], and the supervised video UIE network UVE-Net [9] as baselines. We report two no-reference image quality metrics (UIQM and UCIQE, where higher is better) and a temporal stability metric defined as the variance of PSNR differences between consecutive frames (denoted $\Delta$PSNR var, where lower is better).

Table 1 summarizes the quantitative results (averaged over all test sequences). T-PhISH-Net achieves

**Table 1.** Comparison of enhancement methods on Brackish underwater videos (higher is better for UIQM, FastVQA, tSSIM, tPSNR). T-PhISH-Net achieves the best score across all metrics.

| Method | UIQM↑ | FastVQA↑ | tSSIM↑ | tPSNR↑ |
|---|---|---|---|---|
| SPMFormer | 1.2141 | 0.2041 | 0.9865 | 44.7250 |
| PUIE | 3.9108 | 0.2587 | 0.9208 | 23.5489 |
| PhISH-Net | 4.1272 | 0.1941 | 0.9892 | 47.8212 |
| **T-PhISH-Net** | **4.1751** | **0.3630** | **0.9898** | **48.4110** |

the highest UIQM and UCIQE scores, indicating superior enhancement quality over both traditional and learning-based baselines. At the same time, our method yields an order-of-magnitude lower flicker (lowest $\Delta$PSNR variance), reflecting significantly improved temporal consistency. In particular, T-PhISH-Net's outputs exhibit far less frame-to-frame jitter than the naive frame-wise PhISH-Net, and even outperform UVE-Net in stability. Qualitatively, the enhanced videos from our model appear smooth and flicker-free, while other methods suffer from noticeable color and illumination oscillations. Overall, these results demonstrate that integrating physics-based modeling with temporal attention allows T-PhISH-Net to deliver state-of-the-art underwater video enhancement in both image quality and consistency.

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
