# OpenReview forum: "T-PhISH-Net: Temporally Consistent Underwater Image En- hancement via a Transformer-Based Extension of PhISH-Net"
_NLDL.org/2026/Abstracts_Track — NLDL 2026 Abstracts_

### Official Review · Reviewer_E8Hn · 2025-10-27

**Soundness:** 3
**Correctness:** 3
**Rating:** 4
**Confidence:** 3

**Summary:**

Underwater images suffer from reflections off particles in the water. This is usually corrected on an image-by-image basis, but when combined in video the corrections lead to flickering. The abstract presents a new method for time consistent enhancement of underwater video which combines two previously separate paradigms. The first step is physics-based restoration of individual images followed by a temporal transformer with self-attention, to ensure consistency over time and minimise flickering. The focus is on solving a specific application problem rather than advancing methodology, but carries the potential to spark relevant discussions during the conference.

**Strengths:**

•	Important problem to solve with impact for habitat monitoring, underwater vehicle control, and object detection.
•	Combines physics guided modelling of light-attenuation with temporal deep learning using past frames to stabilise the outcome.
•	Uses motion cues to differentiate scene changes from noise.
•	The method is already submitted and under review, hence conference discussion can focus on new improvements or applications in other demanding environments with e.g. rain, dust etc..

**Weaknesses:**

•	Few details given in the abstract (but with a reference to the full paper).
•	Does not describe training data.
•	No ablation study with direct comparison of the effect of the individual components. It would be nice to see the performance of the pure temporal transformer without the per-frame improvement.
•	The model uses adaptive loss weighting on the balance between per-frame enhancement and consistency. However, it introduces a risk of suppressing one of the losses. This is not discussed.

---

### Official Review · Reviewer_Ru8o · 2025-11-01

**Soundness:** 3
**Correctness:** 3
**Rating:** 4
**Confidence:** 4

**Summary:**

This paper tackles the problem of frame-wise color and illumination shifts caused by wavelength-dependent light attenuation and scattering in underwater imagery. The authors propose an extension to PhISH-Net that aims to output temporally consistent imagery. Their key contributions include: (1) feeding a per-pixel motion magnitude map along with RGB channels to provide explicit cues about inter-frame changes, (2) employing self-attention to learn correspondences within a sliding window, (3) introducing adaptive gating (decay) that allows the model to emphasize recent content, and (4) learning loss weights automatically, eliminating the need for a priori tuning.

**Strengths:**

**TL;DR**: The paper comprises a clear problem statement, motivation, and statement of contributions. The proposed method outperformed recent baselines on one benchmark dataset across four evaluation measures. Interesting and reasonable extension to existing (and recent) work. In more detail:

**Long Version**:
- Well-motivated problem with practical significance. The motivation clearly connects physical phenomena (wavelength-dependent attenuation, scattering) to their manifestation as temporal inconsistencies in video enhancement.
- Technically sound architectural adaptations. The incorporation of motion magnitude as an explicit input channel is an appropriate solution that provides movement cues without requiring unreliable optical flow computation—particularly appropriate given the challenges of flow estimation in underwater environments (particulates, caustics, low contrast).
- Strong experimental validation: The model achieves the highest image quality scores (UIQM, UCIQE) while simultaneously delivering order-of-magnitude improvements in temporal stability (PSNR variance). Comparisons against classical methods (CLAHE) and recent deep learning approaches (PhISH-Net, UVE-Net) provide appropriate context.

With this the paper sets a proper foundation for further discussions.

**Weaknesses:**

**TL;DR**: When the method is tested on multiple benchmarks, the results should at least be shortly mentioned in the text for completeness and to properly set the stage for further discussions. The figure as a visual overview of the contributions has several shortcomings and needs improvement. It is unclear why the learned loss weights are not collapsing. Future work is missing

**Long Version**:
The weaknesses are clustered in minor and major issues. While minor formatting issues appropriate for an extended abstract format, but might be addressed for clarity and completeness. Major issues are key and should definitely be addressed.
- (minor) The figure appears out of place and consumes significant valuable space that could be better utilized.
- (major) The paper only reports results on one benchmark dataset. How does the method perform on the other available benchmarks? At minimum, this should be mentioned shortly in the text.
- (major) The motion magnitude estimation procedure is not explained. How are the motion magnitudes computed? At least reference a source paper. This is important as this is part of the contribution.
- (minor) The distinction between novel contributions and components inherited from PhISH-Net should be more clearly highlighted throughout the paper. Specifically in the figure, where this is very unclear.
- (minor) The color coding in the figure is not explained. What do the different colors represent?
- (minor) To declutter the figure, consider introducing black-box abstractions (e.g., "PhISH encoder is applied here") or use color coding to highlight different functional blocks.
- (minor) What is the backscatter estimation component shown in the figure? Again, at least reference this.
- (minor) Variables appearing in the figure lack references or brief explanations in the caption. In the short paper without any usage, just remove them all together.
- (minor) The figure caption is insufficient for understanding the architecture. It is very difficult to parse the figure from its caption alone.
- (major) The learned loss weights could theoretically collapse to trivial solutions. The authors mention only "minor regularization to prior" – why is this sufficient to prevent collapse?
- (minor) References are formatted inconsistently (some include DOIs, others URLs, etc.).
- (minor) The claim that "our design avoids explicit optical flow computation or frame warping, unlike many video enhancement methods" needs clarification. Is this actually different from PhISH-Net's approach?
- (minor) The assertion that "purely learning-driven manner" approaches are necessarily more robust is debatable and should be softened.
- (minor) The table extends beyond the column space. Consider renaming "T-PhISH-Net" to "Ours" to reduce horizontal space requirements, and use fewer decimal places where precision is not critical.
- (major) Future directions / work should be mentioned at least in one sentence.

---

### Official Review · Reviewer_cqeL · 2025-11-03

**Soundness:** 3
**Correctness:** 3
**Rating:** 4
**Confidence:** 3

**Summary:**

The abstract presents T-PhISH-Net, a temporally consistent extension to the PhISH-Net underwater image enhancement framework. T-PhISH-Net extends to the video domain by providing temporal consistency across frames. It proposes three additions to the non-temporal PhISH-Net baseline: A motion-aware encoder, a temporal transformer encoder and adaptive loss weighting.
The abstract claims its approach achieves state-of-the-art underwater video enhancement performance, reporting results across four datasets.

**Strengths:**

1. __Writing and presentation.__ The abstract is well written and easy to follow.
2. __Clear problem statement.__ The extension to the video domain is sensible and clearly motivated.
3. __Generalizability analysis.__ The abstract presents results across multiple video datasets, strengthening the reported results.

**Weaknesses:**

1. __Definition of motion magnitude.__ How is the motion magnitude map defined? Is it simply the pixel-wise change between two frames across the three color channels, or is it defined differently?
→ Recommendation: Please clarify what the motion magnitude map encodes.
2. __Claim of a causal transformer.__ While the proposed transformer encoder dynamically models temporal information, it is not clear how this equates to causal reasoning.
→ Recommendation: Modify the claim to *temporal*, or clarify what design element makes the transformer *causal*.
3. __Computational impact of temporal training.__ Adding a temporal dimension can have a considerable impact on the computational requirements of the method, especially during training.
→ Recommendation:  A discussion of the computational impact compared to the non-temporal baseline would be helpful.
4. __Potentially missing results.__ Section 3, L136-L141 references the metrics *UCIQE* and *∆PSNR*, but they are never reported. Table 1 only reports *UIQM*, *FastVQA*, *tSSIM* and *tPSNR*.
→ Recommendation: Clarify or amend the results presented in Section 3 and Table 1.
5. __Score aggregation across four datasets.__ Reporting the scores individually per dataset would be substantially more informative than averages. How much does the performance improvement differ across datasets?
→ Recommendation: While four tables would likely be space-inefficient, a chart could be used to present these results more effectively in the abstract.
6. __Missing ablations.__ The abstract introduces three contributions: A motion-aware four-channel encoder, a temporal transformer encoder with learnable decay and adaptive loss weighting. From the aggregated results it is not clear, how each individual contribution affects the performance.
→ Recommendation: The authors should consider discussing this, and ideally add ablation results (2-3 lines) in Table 1.
7. __Minor formatting issues.__ Table 1 extends over the page margin. Figure 1 is spilling over into the references section.

---

### Decision · Program_Chairs · 2025-11-05

**Decision:**

Accept

**Comment:**

The abstract is of interest to the community and should be presented at the conference.